# Occupation and Female Breast Cancer Mortality in South Africa: A Case–Control Study

**DOI:** 10.3390/ijerph22121878

**Published:** 2025-12-17

**Authors:** Melitah Motlhale, Hlologelo Ramatsoma, Tsoseletso Maabela, Kerry Wilson, Nisha Naicker

**Affiliations:** 1Epidemiology and Surveillance, National Institute for Occupational Health, National Health Laboratory Service, Johannesburg 2001, South Africa; melitahm@nioh.ac.za (M.M.); hlologelor@nioh.ac.za (H.R.); tsoseletsom@nioh.ac.za (T.M.); kerryw@nioh.ac.za (K.W.); 2School of Public Health, Faculty of Health Sciences, University of the Witwatersrand, Johannesburg 2193, South Africa; 3Department of Environmental Health, University of Johannesburg, Johannesburg 2028, South Africa

**Keywords:** job, breast cancer risk, major occupation, sub-major occupation

## Abstract

**Highlights:**

**Public health relevance—How does this work relate to a public health issue?**
Breast cancer is a major cause of cancer-related deaths in South Africa, and the findings presented in this study illustrate the occupational determinants that contribute to cancer mortality.The study links different occupational groups to breast cancer mortality using national data.

**Public health significance—Why is this work of significance to public health?**
This is the first national study demonstrating that occupation is a potential independent predictor of breast cancer mortality in South Africa after adjusting for key demographic and behavioural factors.The results show elevated mortality risks among specific occupational groups, indicating that workplace factors do in fact play a role in breast cancer outcomes.

**Public health implications—What are the key implications or messages for practitioners, policy makers and/or researchers in public health?**
Occupational information should be incorporated when assessing patients for breast cancer prevention and screening programmes with targeted interventions for high-risk occupational groups.Researchers should prioritise research into occupational exposures and working conditions to provide an evidence base for breast cancer control.

**Abstract:**

Breast cancer is the most frequently diagnosed malignancy among South African women and remains a leading cause of cancer-related death, yet the role of occupation as an independent predictor of mortality has not been evaluated nationally. In this unmatched case–control study using 2011–2019 mortality data, we compared 13,207 breast cancer deaths with 64,849 non-malignant circulatory disease deaths among women aged 30 years and older, classifying usual occupation into major and sub-groups. A multivariable binary logistic regression adjusting for age, year of death, education, province of death and smoking status was conducted. We observed that compared with elementary occupations, breast cancer mortality was significantly higher during 2011–2015 among legislators, senior officials and managers (aMOR = 1.79, 95% CI: 1.36–2.36), clerks (aMOR = 1.75, 95% CI: 1.46–2.11), professionals (aMOR = 1.62, 95% CI: 1.36–1.94), craft and related trades workers (aMOR = 1.55, 95% CI: 1.18–2.05), technicians and associate professionals (aMOR = 1.54, 95% CI: 1.21–1.96), and service workers, shop and market sales workers (aMOR = 1.33, 95% CI: 1.10–1.62), with similar patterns persisting in 2016–2019 where technicians and associate professionals (aMOR = 1.69, 95% CI: 1.44–1.98), legislators, senior officials and managers (aMOR = 1.59, 95% CI: 1.20–2.10), professionals (aMOR = 1.47, 95% CI: 1.23–1.75), clerks (aMOR = 1.43, 95% CI: 1.24–1.65), and service workers (aMOR = 1.34, 95% CI: 1.12–1.61) again showed elevated odds. The sub-occupation analyses for 2011–2015 identified strikingly high risks among building and related trades workers excluding electricians (aMOR = 8.01, 95% CI: 3.06–20.96), legal, social and cultural professionals (aMOR = 3.32, 95% CI: 2.18–5.04), and business and administration professionals (aMOR = 2.18, 95% CI: 1.60–2.97). The results underscore occupation as an essential determinant of breast cancer mortality, highlighting the need for targeted prevention and screening strategies in workers.

## 1. Introduction

Among women, there are approximately 2.3 million new breast cancer cases diagnosed annually, and this represents 25% of all malignancies [1]. In 2020, globally, breast cancer accounted for a mortality count of approximately 685,000 among females [2]. Breast cancer is considered the most prevalent cancer affecting women worldwide [3]. While in Sub-Saharan Africa, it was the most diagnosed type of cancer and ranked as the second most common cancer contributing to mortality [4]. In South Africa, breast cancer is the most prevalent cancer, with an incidence rate of 52.6 and a mortality rate of 16.1 per 100,000 of the population [5].

The number of breast cancer cases is expected to double by 2050 [6]. Globally, more than half of breast cancer diagnoses, and two out of three cancer mortalities, took place in low- to middle-income countries in 2020, although previously, breast cancer was regarded as a disease more common in high-income countries [2]. Among African countries, this is a result of limited resources allocated to healthcare screening, funding challenges, and little to no knowledge and other cultural influences related to breast cancer [7].

In South Africa, the majority (63.4%) of Black women were diagnosed in the later stages (III, IV) than early stages (I, II) of breast cancer [8]. Hence, early detection of breast cancer has been highlighted as a need, due to the high rate of late-stage disease detection, lack of diagnostic and therapy options, which resulted in low survival [4,9]. In addition to the lack of possible healthcare utilisation, various occupational and environmental factors play a significant role in the development of breast cancer [10,11]. In the workplace, exposure to occupational agents such as ionising and non-ionising radiation, low levels of melatonin among night shift workers, and chemicals such as pesticides were associated with the development of breast cancer [12].

Although an association between breast cancer and occupational types is still unknown, automotive plastics manufacturing, food canning, metalworking, agricultural work with pesticide exposure, water and air transportation, and other production sectors were linked to a high risk of breast cancer in South Africa, Japan, and Taiwan [13,14,15]. Office-based roles due to minimal physical activity and prolonged sitting during work-hours, which possibly increased oestrogen levels throughout the menstrual cycle and thus could be regarded as risk factors for breast cancer [15].

In contrast, women who mostly held positions that required working outdoors, such as gardening, farming, and woodworking, showed a reduced risk of developing breast cancer [14]. While some occupations operate on regular working hours, night shift work remains widely used in many sectors. According to the International Agency for Research on Cancer (IARC) and the World Health Organisation (WHO) Monographs, night shift jobs typical in hospitals, police services, firefighting, power stations, and transportation were likely considered as breast carcinogens due to exposures such as ionising radiation, unnatural light, and circadian disruptions [16].

Previous studies have identified several risk factors associated with the onset of breast cancer. However, to date, no study in South Africa has evaluated the role of occupation on breast cancer mortality. The purpose of the study was to investigate the association between occupational category and breast cancer mortality among South African women by analysing nationwide mortality records to identify high-risk occupations and inform targeted prevention and screening strategies.

## 2. Materials and Methods

### 2.1. Study Design and Population

We conducted an unmatched case–control study using South African mortality data from 2011 to 2019 from Statistics South Africa (StatsSA). All females aged 30 and older whose underlying cause of death was breast cancer (International Classification of Diseases [ICD-10] C50) served as cases. At the same time, controls were all females whose underlying cause of death was non-malignant circulatory diseases (ICD-10 I00–I99), excluding cerebrovascular (I60–I69) and hypertensive (I10–I15) diseases to minimise overlap with occupational exposures.

All analyses were restricted to women aged 30 years and older with a recorded major occupation. We selected individuals aged 30 years and older to allow sufficient latency for occupational exposures. Given that the statutory working age in South Africa is 18 years, this choice provided an expected minimum of 12 years of potential occupational exposure. Exclusion criteria included individuals with incomplete major occupation records or those younger than 30 years.

### 2.2. Data Sources and Management

The underlying cause of death was captured from death certificates submitted to Statistics South Africa by the Department of Home Affairs, South Africa, between 2011 and 2019. Currently, only data up to 2019 is available for public use. Death certificates are required for all deaths in South Africa. Usual occupation, as reported on death certificates, was classified by StatsSA into ten major and 42 sub-major occupation groups based on the South African Standard Classification of Occupations (SASCO) [17] for 2011–2015; for 2016–2019, only the ten major groups were available in the publicly accessible dataset. The usual occupation denotes the type of work the deceased performed for the majority of their working life. We used the nine major occupation groups in both periods and 41 sub-major occupations in the 2011–2015 period for analysis. Categories for ‘occupations unspecified’ and ‘not economically active’ were excluded because they contain unemployed or unclassified individuals. The nine major occupation categories used were: (1) elementary occupations, (2) clerical support workers, (3) legislators, senior officials and managers, (4) professionals, (5) plant, machine operators, and assemblers, (6) service workers, shop and market sales workers, (7) skilled agricultural, forestry and fishery workers, (8) technicians and associate professionals, (9) craft and related trade workers. The forty-one sub-major occupation groups are provided in the Statistics South Africa SASCO 2012 [17]. This standard was based on the International Standard Classification of Occupations (ISCO-08) standardised classification of occupations.

### 2.3. Statistical Analysis

Participants’ demographic characteristics were summarised using frequencies and percentages disaggregated by case or control status. Pearson’s chi-square test was used to assess differences between characteristics of cases and controls. We report effect-size measures using Cramér’s V to quantify the magnitude of differences between cases and controls because large samples can produce statistically significant *p*-values for trivially minor differences [18]. Unspecified and unknown observations were treated as missing in all analyses. Associations between occupational groups and breast cancer mortality were estimated using unconditional multivariable binary logistic regression, adjusting for age, year of death, education, province of death, and smoking status. Models were fitted using maximum likelihood estimation with robust standard errors to account for potential heteroskedasticity and mild misspecification. Model fit was evaluated with the Hosmer–Lemeshow goodness-of-fit test. Results are presented as adjusted mortality odds ratios (aMORs) with 95% confidence intervals (95% CI). Statistical significance was set at *p* < 0.05. All analyses were conducted using Stata version 19.5 (StataCorp LLC, College Station, TX, USA).

## 3. Results

The analysis included 78,056 females over 30 years of age, comprising 13,207 breast cancer mortality cases (16.9%) and 64,849 non-malignant circulatory disease controls (83.1%). Socio-demographic characteristics of the cases and controls are summarised in Table 1. The distributions by year of death were highly similar between cases and controls, with the majority of events occurring in the period 2017–2019. Although the differences in year of death distribution were statistically significant (*p* < 0.001), the small effect size (Cramér’s V = 0.06) indicates that these temporal differences are not substantively important. In contrast, the age group showed marked differences between controls and cases, highlighted by a Cramér’s V of 0.21 (*p* < 0.001). Controls were disproportionately older, with 54.5% (n = 35,299) in the 70+ years category compared to only 29.3% (n = 3873) of cases. Conversely, cases were significantly more concentrated in the middle-aged bands (40–54 years: 29.4% of cases vs. 14.1% of controls). Differences were noted in educational attainment between controls and cases (Cramér’s V = 0.17, *p* < 0.001). While secondary education was the modal category for both groups (controls: n = 15,057 [43.2%] and cases: 3914 [56.6%]), cases exhibited a notably higher proportion with tertiary education (12.4% vs. 5.7%) and a lower proportion in the none category (7.5% vs. 17.3%) compared to controls. The distribution by province of death occurrence showed small geographic shifts (Cramér’s V = 0.12, *p* < 0.001), such as a higher proportion of cases in the Western Cape (21.1% vs. 13.1% of controls) and a lower proportion in KwaZulu-Natal (13.8% vs. 21.9% of controls). Lastly, smoking status was nearly identical between groups, with 15.1% of cases and 16.3% of controls reporting ‘Yes’ to smoking. Despite a statistically significant result (*p* = 0.005), the association was negligible (Cramér’s V = 0.01), indicating that smoking status does not substantially differ between the case and control groups.

In multivariable binary logistic regression models, several major occupational groups had significantly higher odds of breast cancer mortality compared to elementary occupations (Figure 1). For the period from 2011 to 2015, the adjusted model retained 10,851 observations, comprising 2150 (19.8%) cases and 8701 (80.2%) controls. For the period from 2016 to 2019, the adjusted model retained 30,302 observations, comprising 4667 (15.4%) cases and 25,635 (84.6%) controls. Model fit indicated no evidence of poor fit for 2011–2015 (*p* = 0.126) and 2016–2019 (*p* = 0.187). The 2011 to 2015 period showed broadly similar patterns to those observed in 2016 to 2019. For 2011–2015, the highest relative odds were observed among legislators, senior officials and managers (aMOR = 1.79, 95% CI: 1.36–2.36, *p* < 0.001), clerks (aMOR = 1.75, 95% CI: 1.46–2.11, *p* < 0.001), professionals (aMOR = 1.62, 95% CI: 1.36–1.94, *p* < 0.001), craft and related trades workers (aMOR = 1.55, 95% CI: 1.18–2.05, *p* = 0.002), technicians and associate professionals (aMOR = 1.54, 95% CI: 1.21–1.96, *p* < 0.001), and service workers, shop and market sales workers (aMOR = 1.33, 95% CI: 1.10–1.62, *p* = 0.003). In 2016–2019, the pattern of elevated risk persisted for higher-skilled non-manual occupations, with significantly higher odds among technicians and associate professionals (aMOR = 1.69, 95% CI: 1.44–1.98, *p* < 0.001), legislators, senior officials and managers (aMOR = 1.59, 95% CI: 1.20–2.10, *p* = 0.001), professionals (aMOR = 1.47, 95% CI: 1.23–1.75, *p* < 0.001), clerks (aMOR = 1.43, 95% CI: 1.24–1.65, *p* < 0.001), and service workers, shop and market sales workers (aMOR = 1.34, 95% CI: 1.12–1.61, *p* = 0.002). In contrast, plant and machine operators and assemblers had slightly lower odds of breast cancer mortality compared to those in elementary occupations (aMOR = 0.92, 95% CI: 0.85–1.00, *p* = 0.041).

For the 2011 to 2015 period, data on sub-major occupations were available (Table 2). The adjusted model retained 10,846 observations, comprising 2150 (19.8%) cases and 8696 (80.2%) controls. Model fit was assessed using the Hosmer–Lemeshow goodness-of-fit test, which indicated no evidence of poor fit (*p* = 0.147). This has been added to the manuscript and highlighted accordingly to improve transparency. The adjusted mortality odds ratios (aMORs) for female breast cancer mortality across sub-major occupations showed several groups with significantly higher odds compared to refuse workers and other elementary workers (the reference group). Specifically, administrative and commercial managers had higher odds of breast cancer mortality (aMOR 2.18; 95% CI: 1.24–3.82). Hospitality, retail and other services managers also showed elevated odds (aMOR 1.84; 95% CI: 1.25–2.70). Similarly, teaching professionals had 1.56 times greater odds (95% CI: 1.19–2.06). Business and administration professionals demonstrated approximately twofold higher odds (aMOR 2.18; 95% CI: 1.60–2.97). The largest increase among these professional groups was observed for legal, social and cultural professionals (aMOR 3.32; 95% CI: 2.18–5.04). Moving to associate professional and clerical roles, business and administration associate professionals (aMOR 1.68; 95% CI: 1.20–2.35) and general and keyboard clerks (aMOR 1.77; 95% CI: 1.38–2.27) were at higher odds of breast cancer mortality. Customer service clerks had about twice the odds (aMOR 2.07; 95% CI: 1.40–3.05), and sales workers also had increased odds (aMOR 1.57; 95% CI: 1.12–2.19). Finally, building and related trades workers excluding electricians had the most substantial association with breast cancer mortality, with markedly higher odds (aMOR 8.01; 95% CI: 3.06–20.96).

Notable non-significantly increased odds were seen for Food processing, woodworking, garment and other craft and related trades workers, labourers in mining, construction, manufacturing and transport (aMOR 1.64, 95% CI: 0.99; 2.70), Science and engineering professionals (aMOR 1.97, 95% CI: 0.99; 3.92).

## 4. Discussion

In this study, we investigated the association between occupational category and breast cancer mortality among South African women by analysing nationwide mortality records to identify high-risk occupations and inform targeted prevention and screening strategies. Our findings show that breast cancer was significantly higher in numerous occupational groups, particularly technicians, associate professionals, managers, and professionals—across both study periods, with even stronger associations for specific skilled trades, professional, and clerical sub-occupations in 2011–2015. Based on the demographics of our study, controls were disproportionately older due to the late mortality of circulatory diseases compared to breast cancer, with 54.5% (n = 35,299) in the 70+ years category for controls compared to only 29.3% (n = 3873) of cases.

We found an increased likelihood of breast cancer mortality among the major occupation groups: legislators, senior officials and managers, clerks, professionals, craft and related trades workers, technicians and associate professionals, and service workers, shop and market sales workers. This may be linked to prolonged sitting due to the nature of office-based roles. Previous research has shown that there was an association between sedentary behaviour and an increased rate of breast cancer, although in some studies, no association was found [19]. Sedentary behaviour is typical among various occupations, with around 7.7 h of sitting time and with possibilities of this increasing over time [20].

Among women who had previously worked in the agricultural sector, results showed a high risk of breast cancer, which may have resulted from occupational carcinogens such as exposure to pesticides both directly and indirectly (OR = 2.91; 95% CI 1.51–5.60) [10]. In a study conducted in Morocco, a higher breast cancer risk was also found among agricultural workers (OR = 1.36; 95% CI −1.03–1.82) [10]. Differences in our findings and those in other studies may be due to different occupational exposures in various agricultural sectors.

In our study, an elevated risk of breast cancer mortality was observed among professionals in teaching, business and administration, and information and communication technology. Similar associations have been reported in previous research; however, it has been suggested that these findings may be influenced by a considerable number of confounding factors that were not adequately controlled [3]. Similarly to previous findings, an association found in our study among clerical support workers, such as general and keyboard clerks, and customer service clerks, was reported by another study among clerical workers in Japan [14].

Additionally, we found a significant increase in breast cancer mortality among sales workers in comparison to other service and sales workers in the sub-occupation category. Furthermore, among the craft and related trade workers, those in the building and related trades, excluding electricians, showed a significantly higher risk of breast cancer. In another study, craft and related trade workers showed a slightly higher risk of breast cancer (OR = 1.17; 95% CI 0.32–4.26), although among those who worked for a duration of 5 years in a specific occupation, there was no significant risk (OR = 0.98; 95% CI 0.67–1.45) [10]. However, the low risk may be due to recall bias in reporting occupational exposures after a longer working period.

Nonetheless, MORs were calculated where data on occupation or industry were available; this does not reflect the length of employment. We adjusted for education level in our analyses, thus minimising bias due to health knowledge between occupation groups. Additionally, due to the use of secondary Southern African mortality data from Statistics South Africa (StatsSA), information bias may exist. The availability of minor occupation data for the period 2016–2019, which are occupation sub-categories of the 41 sub-major occupations, would have added more value to the analysis. However, StatsSA data is representative of the overall population and various occupations as categorised by the South African Standard Classification of Occupations (SASCO). This study’s strength lies in its use of nationally representative mortality data, which enhances the generalisability of the findings. Additionally, our results are consistent with several studies on occupation and breast cancer; however, our study is the first study to evaluate occupation as an independent predictor of breast cancer mortality in South Africa.

The limitations of such data have been discussed in detail elsewhere [21]. Specific to this study, information on usual occupation was recorded in only 16.3% of death notifications, despite an employment rate of approximately 24.8% and 28.4% between 2011 and 2019 [22]. This discrepancy suggests that mortality patterns may vary between employed and non-employed individuals. To minimise the risk of misclassification, we excluded records lacking occupational data from our analysis. The completeness of the data is an important limitation, as substantial missing values for key demographic variables, such as educational attainment and smoking status, may introduce bias or residual confounding. However, the missing proportions differed by less than five percentage points between cases and controls, which suggests that the extent of bias due to differential missingness is likely limited. While these gaps restrict the ability to fully adjust for known risk factors, the large sample size and use of nationally representative mortality data still provide valuable insights into occupational patterns of breast cancer mortality in South Africa. A key limitation, however, is that reproductive variables were not available for adjustment. These include use of hormonal contraceptives, parity, age at first childbirth, breastfeeding history, age at menarche, and age at menopause, all of which are known to influence breast cancer risk. The absence of these covariates may have introduced residual confounding, underscoring the need for future studies to integrate reproductive and occupational data for more comprehensive risk assessment.

## 5. Conclusions

Breast cancer mortality in South African women was significantly higher in several occupational groups—most notably technicians and associate professionals, managers, and professionals—across study periods, with even stronger associations for certain skilled trades, professional, and clerical sub-occupations in 2011–2015. These patterns persisted after adjustment for key socio-demographic factors, suggesting occupation as an independent risk factor. The diversity of high-risk roles points to multiple possible exposures, from chemical agents to sedentary work and potentially psychosocial stress. As the first national study on this topic, the findings support the need for detailed occupational exposure assessment, prospective designs, and targeted screening and surveillance for high-risk occupational groups.

## Figures and Tables

**Figure 1 ijerph-22-01878-f001:**
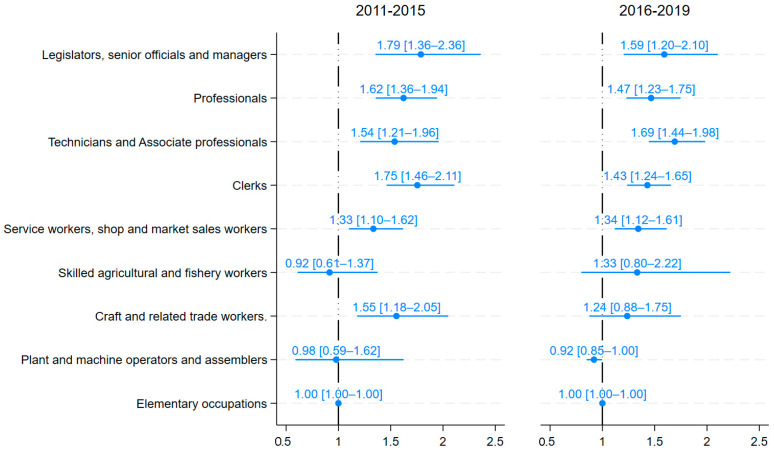
Adjusted female breast cancer mortality odds ratios and 95% confidence intervals, for 2011–2015 and 2016–2019, by major occupation. Adjusted for age, education, year of death, province of death, and smoking status. Reference group: elementary occupations.

**Table 1 ijerph-22-01878-t001:** Socio-demographic characteristics of cases and controls.

	Controls	Cases		
	(N = 64,849)	(N = 13,207)	Cramér’s V	*p*-Value
**Year of death**			0.06	<0.001
2011	2228 (3.4%)	476 (3.6%)		
2012	2025 (3.1%)	520 (3.9%)		
2013	2012 (3.1%)	492 (3.7%)		
2014	1984 (3.1%)	576 (4.4%)		
2015	2104 (3.2%)	581 (4.4%)		
2016	2847 (4.4%)	743 (5.6%)		
2017	17,824 (27.5%)	3199 (24.2%)		
2018	17,816 (27.5%)	3207 (24.3%)		
2019	16,009 (24.7%)	3413 (25.8%)		
**Age group**			0.21	<0.001
30–34 years	1655 (2.6%)	378 (2.9%)		
35–39 years	1896 (2.9%)	725 (5.5%)		
40–44 years	2336 (3.6%)	1076 (8.2%)		
45–49 years	2956 (4.6%)	1319 (10.0%)		
50–54 years	3804 (5.9%)	1484 (11.2%)		
55–59 years	4683 (7.2%)	1537 (11.6%)		
60–64 years	5790 (8.9%)	1450 (11.0%)		
65–69 years	6345 (9.8%)	1360 (10.3%)		
70+ years	35,299 (54.5%)	3873 (29.3%)		
Missing ^a^	85 (.%)	5 (.%)		
**Educational attainment**			0.17	<0.001
None	6012 (17.3%)	517 (7.5%)		
Primary education	11,787 (33.8%)	1625 (23.5%)		
Secondary education	15,057 (43.2%)	3914 (56.6%)		
Tertiary education	1977 (5.7%)	857 (12.4%)		
Missing ^a^	30,016 (.%)	6294 (.%)		
**Province of death occurrence**			0.12	<0.001
Western Cape	8329 (13.1%)	2729 (21.1%)		
Eastern Cape	8228 (12.9%)	1484 (11.5%)		
Northern Cape	2268 (3.6%)	453 (3.5%)		
Free State	4455 (7.0%)	753 (5.8%)		
KwaZulu-Natal	13,954 (21.9%)	1782 (13.8%)		
North West	3838 (6.0%)	785 (6.1%)		
Gauteng	14,851 (23.3%)	3464 (26.7%)		
Mpumalanga	4237 (6.7%)	562 (4.3%)		
Limpopo	3490 (5.5%)	944 (7.3%)		
Missing ^a^	1199 (.%)	251 (.%)		
**Smoking status of the deceased**			0.01	0.005
Yes	6999 (16.3%)	1229 (15.1%)		
No	35,944 (83.7%)	6934 (84.9%)		
Missing ^a^	21,906 (.%)	5044 (.%)		

^a^ The missing proportions differed by less than five percentage points between cases and controls.

**Table 2 ijerph-22-01878-t002:** Adjusted odds ratios and 95% CI for female breast cancer mortality by sub-major occupation from 2011–2015.

Sub-Occupation	Adjusted Odds Ratios (95% CI)
Refuse workers and other elementary workers	1 (ref)
**Legislators, senior officials and managers**	
Chief executives, senior officials and legislators	1.82 (0.92; 3.63)
Administrative and commercial managers	2.18 (1.24; 3.82) *
Production and specialised services managers	1.29 (0.56; 3.00)
Hospitality, retail and other services managers	1.84 (1.25; 2.70) *
**Professionals**	
Science and engineering professionals	1.97 (0.99; 3.92)
Health professionals	1.28 (0.97; 1.70)
Teaching professionals	1.56 (1.19; 2.06) *
Business and administration professionals	2.18 (1.60; 2.97) *
Information and Communication Technology professionals	2.12 (0.46; 9.71)
Legal, social and cultural professionals	3.32 (2.18; 5.04) *
**Technicians and Associate Professionals**	
Science and engineering associate professionals	1.35 (0.62; 2.91)
Health associate professionals	1.48 (0.88; 2.47)
Business and administration associate professionals	1.68 (1.20; 2.35) *
Legal, social, cultural and related associate professionals	1.30 (0.70; 2.40)
Information and communications technicians	4.13 (0.81; 21.02)
**Clerical Support Workers**	
General and keyboard clerks	1.77 (1.38; 2.27) *
Customer service clerks	2.07 (1.40; 3.05) *
Numerical and material recording clerks	1.35 (0.41; 4.42)
Other clerical support workers	2.12 (0.98; 4.57)
**Service and Sales Workers**	
Personal services workers	1.25 (0.92; 1.69)
Sales workers	1.57 (1.12; 2.19) *
Personal care workers	1.53 (0.80; 2.92)
Protective service workers and armed forces occupations	1.27 (0.80; 2.02)
**Skilled Agricultural, Forestry, Fishery and Hunting Workers**	
Market-oriented skilled agricultural workers	1.00 (0.21; 4.81)
Market-oriented skilled forestry, fishery and hunting workers	0.87 (0.52; 1.47)
Subsistence farmers, fishers, hunters and gatherers	1.07 (0.50; 2.28)
**Craft and Related Trades Workers**	
Building and related trades workers, excluding electricians	8.01 (3.06; 20.96) *
Metal, machinery and related trades workers	1.25 (0.80; 1.93)
Handicraft and printing workers	2.15 (0.94; 4.96)
Food processing, woodworking, garment and other craft and related trades workers	1.56 (1.00; 2.42)
**Plant and Machine Operators and Assemblers**	
Stationary plant and machine operators	0.73 (0.38; 1.40)
Assemblers	4.90 (0.61; 39.15)
Drivers and mobile plant operators	2.07 (0.80; 5.35)
**Elementary Occupations**	
Cleaners and helpers	1.01 (0.82; 1.25)
Agricultural, forestry and fishery labourers	0.83 (0.46; 1.49)
Labourers in mining, construction, manufacturing and transport	1.64 (0.99; 2.70)
Food preparation assistants	3.19 (0.82; 12.41)
Street and related sales and service workers	0.59 (0.17; 2.07)

* *p* < 0.05; CI: Confidence intervals; ref: reference. Adjusted for age, education, year of death, province of death, and smoking status.

## Data Availability

The data supporting this study’s findings are available upon request from N.N.

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
