# Peer review of "Occupation and Female Breast Cancer Mortality in South Africa: A Case–Control Study"

_ijerph, 2025, doi:10.3390/ijerph22121878_

Round 1

Reviewer 1 Report

Comments and Suggestions for Authors

This article examines breast cancer mortality in South African women in the context of occupation. The findings highlight that occupation is a significant determinant of breast cancer mortality. This highlights the need for targeted prevention and screening strategies for workers. This is the first study of its kind conducted in South Africa. The introduction introduces the topic addressed by the researchers. Materials and methods are well described. Results are presented in descriptive, tabular, and graphical formats. The discussion describes the most important results obtained and compares them with other findings. Conclusions are drawn based on the obtained results.

I have a few minor comments that will improve the quality of the article.

Introduction: It's worth writing at the end of the Introduction what the purpose of the study was.

Materials and Methods: The authors of this article specified inclusion criteria for the case-control study. It is worthwhile to add exclusion criteria.

Discussion: Please add a paragraph describing the strengths and weaknesses of the study. It's also worth adding implications for the future.

Line 73: WHO. The abbreviation is missing an explanation.

Line 114: ISCO. The abbreviation is missing an explanation.

Author Response

For research article:

Occupation and Female Breast Cancer Mortality in South Africa: A case–control study

Response to Reviewer 1 Comments

1. Summary

Thank you very much for taking the time to review this manuscript and highlighting its strengths and limitations. Please find detailed responses below, along with the corresponding revisions/corrections highlighted in track changes in the resubmitted files.

2. Questions for General Evaluation

Reviewer’s Evaluation

Response and Revisions

Does the introduction provide sufficient background and include all relevant references?

Yes/

Are all the cited references relevant to the research?

Yes

Is the research design appropriate?

Yes

Are the methods adequately described?

Can be improved                     

Detailed comments have been addressed below

Are the results clearly presented?

Yes

Are the conclusions supported by the results?

Yes

3. Point-by-point response to Comments and Suggestions for Authors

Comments 1: It's worth writing at the end of the Introduction what the purpose of the study was.

Response 1: We initially stated the aim of the study; however, we agree with you that it is essential to clearly state the purpose of the study. We have clearly added this and highlighted it in lines 79-82 of the introduction.

Comments 2: The authors of this article specified inclusion criteria for the case-control study. It is worthwhile to add exclusion criteria.

Response 2: We thank the reviewer for this suggestion and have now explicitly added exclusion criteria to clarify our study participants in lines 96 and 97.

Comments 3: Discussion: Please add a paragraph describing the strengths and weaknesses of the study. It's also worth adding implications for the future.

Response 3: We appreciate this observation. We have clarified the strengths and limitations of the study, as well as its implications for the future. These were added and highlighted between lines 271 to 294.

Comments 4: Line 73: WHO. The abbreviation is missing an explanation.

             Line 114: ISCO. The abbreviation is missing an explanation.

Response 4: Line 73: WHO. The abbreviation is missing an explanation, and Line 114: ISCO. The abbreviation is missing an explanation

4. Response to Comments on the Quality of English Language

Point 1: The English is fine and does not require any improvement.

Response 1: No changes required

 Thank you once again for the review. 

Reviewer 2 Report

Comments and Suggestions for Authors

This manuscript (MS ID: ijerph-3967111) analyzes 9 years of national data (2011–2019) to examine the association between occupational status and breast cancer mortality, while adjusting for demographic variables. I have several comments that require clarification to strengthen the rigor and interpretability of the manuscript:

1. The dataset includes 13,207 cases and 64,849 controls; however, there are substantial missing values for key demographic variables such as educational attainment and smoking status in both groups. The authors should clarify whether the missingness in these variables could introduce bias or misclassification, and discuss the extent to which the findings may be affected.

2. Please report the exact analytical sample size used in each model. Since the adjusted analyses exclude records with missing demographic data, it is essential to specify how many cases and controls were retained in the final models.

3. In Table 2, a single model appears to have been used with “Refuse workers and other elementary workers” as the reference category to estimate AOR for 38 sub-occupational groups. The authors should provide appropriate model performance diagnostics (e.g., sensitivity, specificity, pseudo-R² values, goodness-of-fit statistics) to support the validity of this model.

4. The manuscript lacks a limitations section. The authors are strongly encouraged to discuss key limitations, particularly those associated with the case–control design, missing data, potential residual confounding, and the use of national time-series data. Addressing these issues would significantly improve transparency and strengthen the discussion.

Author Response

For research article:

Occupation and Female Breast Cancer Mortality in South Africa: A case–control study

Response to Reviewer 1 Comments

1. Summary

Thank you very much for taking the time to review this manuscript and highlighting its strengths and limitations. Please find detailed responses below, along with the corresponding revisions/corrections highlighted in track changes in the resubmitted files.

2. Questions for General Evaluation

Reviewer’s Evaluation

Response and Revisions

Does the introduction provide sufficient background and include all relevant references?

Yes/

Is the research design appropriate?

Yes

Are the methods adequately described?

Can be improved                     

Detailed comments have been addressed below

Are the results clearly presented?

Can be improved                     

Detailed comments have been addressed below

Are the conclusions supported by the results?

Yes

3. Point-by-point response to Comments and Suggestions for Authors

Comments 1: The dataset includes 13,207 cases and 64,849 controls; however, there are substantial missing values for key demographic variables such as educational attainment and smoking status in both groups. The authors should clarify whether the missingness in these variables could introduce bias or misclassification, and discuss the extent to which the findings may be affected.

Comments 4: The manuscript lacks a limitations section. The authors are strongly encouraged to discuss key limitations, particularly those associated with the case–control design, missing data, potential residual confounding, and the use of national time-series data. Addressing these issues would significantly improve transparency and strengthen the discussion. The section should be revised to remove descriptive repetition and focus exclusively on the interpretation of the findings.

Response 1&4 Combined: We thank the reviewer for highlighting this issue and have now clarified that, although the completeness of the data is a limitation, the missing proportions differed by less than five percentage points between cases and controls, which mitigates the risk of substantial bias. We have also addressed your comment on the lack of a limitation section, ensuring discussion of recall bias associated with our study design, missing data, potential residual confounding, and the use of national time-series data. These have been addressed and highlighted from lines 263 to 294.

Comments 2: Please report the exact analytical sample size used in each model. Since the adjusted analyses exclude records with missing demographic data, it is essential to specify how many cases and controls were retained in the final models

Response 2: We thank the reviewer for this important suggestion. We have now clarified the analytical sample size retained in each adjusted model, specifying the exact number of cases and controls after excluding records with missing data. These have been highlighted at the beginning of each model’s interpretation.

Comments 3: In Table 2, a single model appears to have been used with “Refuse workers and other elementary workers” as the reference category to estimate AOR for 38 sub-occupational groups. The authors should provide appropriate model performance diagnostics (e.g., sensitivity, specificity, pseudo-R² values, goodness-of-fit statistics) to support the validity of this model.

Response 3: We appreciate the reviewer's suggestion. We have assessed model fit using the Hosmer–Lemeshow goodness‑of‑fit test, which indicated no evidence of poor fit for all three models. This has been added to the manuscript and highlighted accordingly

4. Response to Comments on the Quality of English Language

Point 1: The English is fine and does not require any improvement.

Response 1: No changes required

Reviewer 3 Report

Comments and Suggestions for Authors
  1. Authors appear to state that they used morality data for years from 2011-2019, however when presenting the results, they used data from 2011-2025. The reason for excluding other years is not stated clearly
  2. Many professionals were found to be significantly associated with breast cancer mortality. This is most likely due failure to rule out effect of several confounders. On the other hand, Table 2 which is logistic regression has so many variables, which makes one wonder if at all there was any variable left out on bivariate regression. Data for bivariate regression is missing.
  3. In Table 2, if the intention was to adjusted odds ratio, there should be no need to have a reference category. Hence, the whole analysis in Table 2 appears wrongly done. Hence, the need for statistician expert to verify this analysis

Author Response

For research article:

Occupation and Female Breast Cancer Mortality in South Africa: A case–control study

Response to Reviewer 3 Comments

1. Summary

Thank you very much for taking the time to review this manuscript and highlighting its strengths and limitations. Please find detailed responses below, along with the corresponding revisions/corrections highlighted in track changes in the resubmitted files.

2. Questions for General Evaluation

Reviewer’s Evaluation

Response and Revisions

Does the introduction provide sufficient background and include all relevant references?

Can be improved                     

Detailed comments have been addressed below

Is the research design appropriate?

Can be improved                     

Detailed comments have been addressed below

Are the methods adequately described?

Can be improved                     

Detailed comments have been addressed below

Are the results clearly presented?

Can be improved                     

Detailed comments have been addressed below

Are the conclusions supported by the results?

Are all figures and tables clear and well-presented?              

Can be improved 

Must be improved                   

Detailed comments have been addressed below

Detailed comments have been addressed below

3. Point-by-point response to Comments and Suggestions for Authors

  1. Comments 1: Authors appear to state that they used morality data for years from 2011-2019, however when presenting the results, they used data from 2011-2025. The reason for excluding other years is not stated clearly.

Response 1: Thank you for the review of our manuscript.

In the results section, we have only used data from 2011 to 2019. We used “control find” in our document to see where we might’ve stated 2025, and we cannot find this in our document. Currently, only data up to 2019 is available for public use. We have added this and highlighted it in line 101.

Comments 2: Many professionals were found to be significantly associated with breast cancer mortality. This is most likely due failure to rule out effect of several confounders. On the other hand, Table 2 which is logistic regression has so many variables, which makes one wonder if at all there was any variable left out on bivariate regression. Data for bivariate regression is missing.

Response 2: In our analysis, we did not conduct separate bivariate regressions, as our focus was on adjusted models. These models accounted for the available demographic covariates in the Statistics South Africa dataset, specifically age, year of death, province of death, education, and smoking status. While we acknowledge that reproductive factors such as parity, age at first childbirth, breastfeeding history, and age at menarche/menopause could influence breast cancer risk, these variables were not captured in the dataset, and we have highlighted this limitation in the discussion. From a statistical perspective, it is acceptable to present fully adjusted models without first reporting bivariate associations, as the adjusted models directly estimate the independent effect of each occupational group while controlling for confounding. Importantly, all 42 sub‑major occupational categories were included in the regression models, ensuring that no occupational group was omitted from the adjusted analysis.

Comments 3: In Table 2, if the intention was to adjusted odds ratio, there should be no need to have a reference category. Hence, the whole analysis in Table 2 appears wrongly done. Hence, the need for statistician expert to verify this analysis

Response 3: We thank the reviewer for this comment. In logistic regression, categorical predictors must be parameterised relative to a reference category in order to estimate odds ratios. This applies equally to adjusted models, as the reference group provides the baseline against which all other categories are compared. Our analysis, therefore, followed standard statistical practice, with all 42 sub-major occupational categories included and one category serving as the reference category. We confirm that the modelling approach was correctly implemented and note that the second author, a biostatistician, verified the analysis and results.

4. Response to Comments on the Quality of English Language

Point 1: The English is fine and does not require any improvement.

Response 1: No changes required

Round 2

Reviewer 3 Report

Comments and Suggestions for Authors

Authors have addressed raised concerns sufficiently